# Zn-Enhanced Asp-Rich Antimicrobial Peptides: N-Terminal Coordination by Zn(II) and Cu(II), Which Distinguishes Cu(II) Binding to Different Peptides

**DOI:** 10.3390/ijms22136971

**Published:** 2021-06-28

**Authors:** Adriana Miller, Agnieszka Matera-Witkiewicz, Aleksandra Mikołajczyk, Joanna Wątły, Dean Wilcox, Danuta Witkowska, Magdalena Rowińska-Żyrek

**Affiliations:** 1Faculty of Chemistry, University of Wroclaw, F. Joliot-Curie 14, 50-383 Wroclaw, Poland; adriana.miller@chem.uni.wroc.pl (A.M.); joanna.watly@chem.uni.wroc.pl (J.W.); 2Screening Laboratory of Biological Activity Tests and Collection of Biological Material, Faculty of Pharmacy, Wroclaw Medical University, Borowska 211A, 50-556 Wroclaw, Poland; agnieszka.matera-witkiewicz@umed.wroc.pl (A.M.-W.); aleksandra.mikolajczyk@umed.wroc.pl (A.M.); 3Department of Chemistry, Dartmouth College, 6128 Burke Laboratory, Hanover, NH 03755, USA; dean.e.wilcox@dartmouth.edu; 4Institute of Health Sciences, University of Opole, 68 Katowicka St., 45-060 Opole, Poland

**Keywords:** metal-antimicrobial peptide interactions, thermodynamics, Zn(II) and Cu(II) bioinorganic chemistry

## Abstract

The antimicrobial activity of surfactant-associated anionic peptides (SAAPs), which are isolated from the ovine pulmonary surfactant and are selective against the ovine pathogen *Mannheimia haemolytica*, is strongly enhanced in the presence of Zn(II) ions. Both calorimetry and ITC measurements show that the unique Asp-only peptide SAAP3 (DDDDDDD) and its analogs SAAP2 (GDDDDDD) and SAAP6 (GADDDDD) have a similar micromolar affinity for Zn(II), which binds to the N-terminal amine and Asp carboxylates in a net entropically-driven process. All three peptides also bind Cu(II) with a net entropically-driven process but with higher affinity than they bind Zn(II) and coordination that involves the N-terminal amine and deprotonated amides as the pH increases. The parent SAAP3 binds Cu(II) with the highest affinity; however, as shown with potentiometry and absorption, CD and EPR spectroscopy, Asp residues in the first and/or second positions distinguish Cu(II) binding to SAAP3 and SAAP2 from their binding to SAAP6, decreasing the Cu(II) Lewis acidity and suppressing its square planar amide coordination by two pH units. We also show that these metal ions do not stabilize a membrane disrupting ability nor do they induce the antimicrobial activity of these peptides against a panel of human pathogens.

## 1. Introduction

As the number of drug-resistant pathogens is continuously increasing, novel therapies are being sought intensively. Antimicrobial peptides (AMPs) are a group of molecules from the innate immune system of a wide range of organisms. Because of their natural occurrence, high effectiveness, various mechanisms of action, and low probability for drug resistance, they are considered an excellent starting point for new therapeutics [1,2]. Currently, the AMP database contains over 3200 compounds, most of which are cationic peptides [3]. Peptides from the surfactant-associated anionic peptides (SAAP) family are, unlike most AMPs, anionic, and this unique character, together with an unusual, Asp-rich sequence (Figure 1) makes them particularly interesting from a molecular and therapeutic point of view.

SAAPs were isolated from the ovine pulmonary surfactant, whose main task is to reduce surface tension at the air/liquid interface in alveoli, where they can inhibit bacterial infections [4,5]. Most probably, SAAPs originate from surfactant cationic pro-peptides, where they are an N-terminal charge-neutralizing fragment. During post-translational modification, this fragment is cleaved off and the peptide is activated [6].

SAAP2 (GDDDDDD), SAAP3 (DDDDDDD), and SAAP6 (GADDDDD) (Figure 1) exhibit bactericidal activity, mainly against *Mannheimia haemolytica* [4,5,6], a Gram-negative bacterium that is the major cause of respiratory mortality in cattle, sheep, and goats due to pneumonic mannheimiosis (shipping fever). Healthy animals carry *M. haemolytica* as a nasopharyngeal commensal. However, when cattle are stressed or become infected with respiratory viruses, *M. haemolytica* replicates and is inhaled into the lower part of the respiratory tract, causing severe damage. Infection is treated with antibiotics, and vaccines are used for prevention [7,8]. In rare cases, *M. haemolytica* can also cause septicemia in human infants and immunocompromised adults [9].

Other pathogens, such as *E. coli, P. aeruginosa, K. pneumonia,* and *S. aureus,* are also susceptible to antimicrobial SAAP peptides, but with high minimal inhibitory concentration (MIC) values in the 470 to 1235 μg/mL range [5,6]. The detailed mode of action is still unclear, but organelle distortion without cell membrane disruption was observed, implicating intracellular targeting [5,10]. Of particular interest, is the requirement for Zn(II) to induce bacteria-killing [4]. This finding motivates several questions. Does the binding of Zn(II) cause a structural rearrangement or is it simply due to the extra positive charge carried by this ion? Does the binding of Zn(II) make these peptides potent against common human pathogens, in addition to *M. haemolytica*? Does the Zn(II) allow the peptide to become membrane-disrupting? Finally, does Cu(II), which is also abundant in the ovine respiratory tract, have a similar effect as Zn(II) [11]?

Both zinc and copper are necessary for the proper functioning of many enzymes that are crucial for living processes in prokaryotes and eukaryotes and are targeted by the innate defense with a mechanism known as nutritional immunity [2,12,13]. Since these metal ions are necessary for pathogen survival and virulence, the host binds them, thereby reducing their bioavailability and leading to the pathogen’s impairment and ultimately death [2,13].

Another way to affect a pathogen is by exploiting transition metal toxicity. It has been observed, that the host can generate a local overload of metal ions, which can disrupt a pathogen [13]. Finally, Cu(II) and Zn(II) can sometimes boost a peptides’ antimicrobial activity by changing its secondary structure or charge [2].

Intrigued by the influence of Zn(II) on SAAP activity, we decided to examine the formation, stability, and structure of Zn(II)-SAAP complexes. Due to relatively high Cu(II) concentrations in certain biological systems, and reports that antimicrobial activity of other AMPs increases in the presence of this metal [14,15], Cu(II)-SAAP complexes were also examined.

Metal enhancement of antimicrobial activity was not the only motivation for this study. The sequences of SAAP2, SAAP3, and SAAP6 differ in their first two N-terminal amino acids (GD, DD, GA), with the rest of the sequence consisting solely of (five) Asp residues. Since Cu(II) and Zn(II) often bind at the N-terminus of proteins and peptides, this provides a naturally occurring series to investigate this biological coordination chemistry. To the best of our knowledge, this is the first attempt to characterize the metal complexes of such unique and surprisingly Asp-rich peptides.

To better understand the SAAP bioinorganic chemistry, we investigated the Zn(II) and Cu(II) complexes of SAAP2, SAAP3, and SAAP6 by a variety of analytical methods. Mass spectrometry indicates the stoichiometry of the complexes and also checks the purity of the samples. Potentiometric titrations determine the protonation constants and stability constants for distribution and competition diagrams that describe the stability of the complexes and suggest the mode of coordination. Isothermal titration calorimetry data provided the thermodynamics of complex formation. Spectroscopic measurements indicate the number and type of donor atoms in the Cu(II) complexes, depending on the pH. Liposome experiments confirm that neither the peptides nor their Cu(II) or Zn(II) complexes are membrane-disrupting molecules.

## 2. Results and Discussion

### 2.1. Mass Spectrometry

To determine the complex stoichiometry, MS measurements were carried out.

For the Zn(II)-SAAP2 sample, two peaks are observed. The most intense one (*m/z* = 766.2, *z* = +1) corresponds to the free peptide. The second one (*m/z =* 828.1*, z* = +1) is the Zn(II)-SAAP complex, which is clearly confirmed by comparison of the experimental and simulated spectra (Appendix A). For the Cu(II)-SAAP2 sample, three major signals are detected, corresponding to the free peptide (*m/z* = 766.2, *z* = +1), sodium adduct (*m/z* = 788.2, *z*= +1) and Cu(II) complex (*m/z* = 827.1, *z* = +1) (Appendix A).

For the Zn(II)-SAAP3 sample, the two most intense signals correspond to the free peptide and the Zn(II) complex (*m/z* = 824.2 and 886.1 respectively, *z* = +1) (Appendix A). For the Cu(II)-SAAP3 sample, peaks corresponding to the free peptide (*m/z* = 824.2, *z* = +1), sodium adduct (*m/z* = 846.2, *z* = +1), potassium adduct (*m/z* = 862.2, *z* = +1) and Cu(II) complex (*m/z* = 885.1, *z* = +1) are observed (Appendix A).

For the Zn(II)-SAAP6 sample, three major signals are assigned to the free peptide, (*m/z* = 722.2, *z* = +1), sodium adduct (*m/z* = 744.2, *z* = +1) and Zn(II) complex (*m/z* = 784.1, *z* = +1) (Appendix A). Similarly, three peaks are observed for the Cu(II)-SAAP6 sample. These signals correspond to the free peptide, (*m/z* = 722.2, *z* = +1), sodium adduct (*m/z* = 744.2, *z*= +1) and Cu(II) complex (*m/z* = 783.1, *z* = +1) (Appendix A).

### 2.2. Potentiometric Titration and Spectroscopic Studies

#### 2.2.1. Protonation of the Ligands

Over the pH range 2–11, SAAP2 (GDDDDDD) and SAAP6 (GADDDDD) show six protonation constants and SAAP3 (DDDDDDD) shows seven. All three peptides deprotonate similarly, with deprotonation of the N-terminal amine (LH = 8.74, 8.70, and 8.43, respectively) the first constant and the remaining constants from the aspartic acid residues. Results from the potentiometric studies were presented in Table 1.

#### 2.2.2. Zinc(II) Complexes

Zn(II)-SAAP2 complexes are found in six different protonation states (Table 1). Zn(II) starts to interact with SAAP2 at pH above 4, resulting in a ZnH_2_L form, in which carboxylate side chains are the anchoring sites for the Zn(II). The next form (ZnHL) with p*K*_a_ = 4.86 corresponds to the deprotonation of an aspartic acid residue, which contributes to Zn(II) binding (the p*K*_a_ of this residue in the free peptide is 5.56). The following complex (ZnL), with p*K*_a_
*=* 6.31, dominates over the pH range 6.4–8.3, with a maximum at 7.4. This form includes coordination of the N-terminal amine (the protonation constant in the free peptide is 8.74; a significant difference between the protonation and stability constants indicates coordination of the group). The next three species (ZnH_−1_L, ZnH_−2_L, and ZnH_−3_L), with p*K*_a_ values 8.24, 8.94, and 9.35, respectively, arising from deprotonation of water ligands of the Zn(II)-peptide complex (Appendix A).

For the Zn(II)-SAAP3 complex, seven forms are observed (Table 1). The first is the carboxylate-bound ZnH_3_L, detected at pH around 4.5. For the next complex, ZnH_2_L, with p*K*_a_ = 4.65 (4.98 in the free peptide), the coordination most likely does not change. For ZnHL, which dominates over the pH range 5–6.5, another carboxylate takes part in the Zn(II) binding (p*K*_a_ = 4.92 for the complex and 5.89 for the free peptide). The ZnL complex is the most abundant form at pH 6.5–8.3; here, the N-terminal amine participates in binding (p*K*_a_ = 6.54 for the Zn(II) complex and 8.43 for the free peptide). Similar to the Zn(II)-SAAP2 complexes, the last three species, ZnH_−1_L, ZnH_−2_L, and ZnH_−3_L (p*K*_a_ = 8.34, 8.78 and 9.14, respectively) arise from deprotonation of water ligands that are coordinated to the Zn(II) (Appendix A).

The Zn(II)-SAAP6 complex exists in five different protonated forms (Table 1). Zn(II) starts to bind to this peptide at a pH of around 5, resulting in the ZnHL species. The next complex (ZnL), with p*K*_a_ = 6.05 dominates over the pH range 6.0–8.0 and corresponds to coordination by the N-terminal amine. Also, as in the previous complexes, the last three species (ZnH_−1_L, ZnH_−2_L, and ZnH_−3_L), with p*K*_a_ = 8.01, 8.95, and 9.35, respectively, reflect the deprotonation of water ligands (Appendix A).

#### 2.2.3. Cu(II) Complexes

For the Cu(II)-SAAP complexes, potentiometric, as well as absorption, CD, and EPR spectroscopic, measurements have been carried out.

Cu(II)-SAAP2 shows six complex forms (Table 1). The first acidic one (CuH_4_L) appears at a pH above 3. The next (CuH_2_L) is observed at a pH around 4 and achieves its maximum at about pH 4.5 (Appendix A). The 760 nm absorption maximum that corresponds to this species suggests the coordination of one nitrogen, indicating that the N-terminal amine is the anchoring site in the Cu(II)-SAAP2 complex [17] (Figure 2A).

The next complex (CuHL) comes from deprotonation of an aspartic acid residue; its non-bonding character is suggested by the small difference between the stability and protonation constants for this group (p*K*_a_ = 4.72 and 4.69, respectively). In the CuL form, another aspartic acid deprotonates and, in this case, it may participate in Cu(II) binding (p*K*_a_ = 4.87, compared to a protonation constant of 5.56). The last three species (CuH_−1_L, CuH_−2_L, CuH_−3_L) correspond to the coordination of three amide groups. In the CuH_−1_L complex (p*K*_a_ = 6.02), coordination of the first amide nitrogen begins at pH above 5.5 and dominates over the pH range 6.2–7.8 (Appendix A).

Coordination of the amide nitrogen is confirmed by the absorption spectra, where a blue shift of the maximum (690 →620 nm) at pH 5–7 can be observed for the coordination of two nitrogens [17] (Figure 2A). The EPR spectrum at pH 6.5 (Appendix A) shows overlapped signals for at least two Cu(II) species, most likely with 1N or 2N, or 2N or 3N coordination [18,19,20]. The distribution diagram, Appendix A, supports the overlap of different complex species at this pH. The second amide group (CuH_−2_L; p*K*_a_ = 7.32) begins to bind Cu(II) at a pH around 7 and dominates over the pH range 7.8–9.9 (Appendix A). The absorption spectrum indicates the binding of a second amide (third nitrogen) at pH 8 by a shift of the maximum to 560 nm [17] (Figure 2A). Moreover, the presence of two characteristic signals (Cotton effect) at 500(+) and 590(-) nm in the CD spectrum confirms that a square-planar complex forms at pH 8 (Figure 3A) [19]. The EPR data support these interpretations, showing A_‖_ = 179.0 G and g_‖_ = 2.252, which are characteristic of a three nitrogen donor set [18]. The third amide begins to coordinate the Cu(II) at around pH 9 (CuH_−3_L) and prevails above pH 9.9 (Appendix A). The binding of four nitrogens is also indicated by the absorption spectrum, where the maximum at pH 9–11 moves to 550 nm [17] (Figure 2A). In the CD spectrum, the intensity of the two characteristic peaks at 500(+) and 590(-) nm significantly increases over the pH range 9–11, indicating a rising concentration of the square-planar complex. The EPR parameters (Table 2), specifically the number of super hyperfine lines in the perpendicular region (Appendix A), also strongly support the 4N coordination above pH 9 [18].

Cu(II)-SAAP3 exists in eight protonation forms (Table 1). The first acidic form (CuH_5_L) achieves its maximum at pH of around 3.4 (Appendix A). The absorption spectrum at pH 4 suggests the coordination of one nitrogen, presumably the N-terminal amine [17] (Figure 2B). The next three constants (CuH_2_L, CuHL, CuL) arise from the deprotonation of non-bonding aspartic acid residues and, as in the case of SAAP2, only one of them is involved with Cu(II) binding, as suggested by a difference between the stability and protonation constants (5.33 and 5.89 for the complex and free peptide, respectively) (Table 1). The last three species (CuH_−1_L, CuH_−2_L, CuH_−3_L) involve the coordination of three amides (p*K*_a_ = 5.90, 7.70, and 9.42, respectively). The CuH_−1_L form with one coordinated amide dominates over the pH range 6.0–7.8 (Appendix A). This is supported by the gradual shift of the maximum in the absorption spectrum to shorter wavelengths (650 → 640 nm) over the pH 6–7 range, indicating the coordination of a second nitrogen [17] (Figure 2B). The EPR spectrum at pH 6.5, as in the case of Cu(II)-SAAP2, shows overlapping signals from different complexes, with 1N or 2N and 3N donor sets (Appendix A and Table 2) [18]. The second amide group binds to Cu(II), resulting in a CuH_−2_L species with a maximum at around pH 8.8 (Appendix A) and a corresponding absorption maximum at 580 nm, confirming three nitrogens in the Cu(II) coordination [17] (Figure 2B).

Moreover, the CD spectrum has two bands at 490(+) and 583(-) nm that start to appear at pH 8 and whose intensity gradually increases at pH 9–11, confirming the formation of the square-planar complex (Figure 3B) [19]. The EPR parameters: A_‖_ = 182.0 G and g_‖_ = 2.247 are consistent with 3N coordination (Table 2) [18]. The third amide starts to coordinate at around pH 9 (CuH_−3_L) and this form prevails at pH above 9.7 (Appendix A). These results are confirmed by the 550 nm absorption maximum at pH 9–11, which is consistent with coordination by four nitrogens [17] (Figure 2B) and further confirmed by the EPR spectrum, with A_‖_ = 206.0 G and g_‖_ = 2.201 (Table 2) [18,19].

Cu(II)-SAAP6 has seven complex forms (Table 1), with the first (CuH_4_L) reaching its maximum at around pH 3.5 (Appendix A). The corresponding absorption maximum at 730 nm indicates the coordination of one nitrogen over the pH range 4–5, which implicates the N-terminal amine as an anchoring site, as found with SAAP2 and SAAP3 [17] (Figure 2C). The next three species (CuH_2_L, CuHL, CuL) arise from deprotonation of non-bonding aspartic acid residues as there are only minor differences between the complex stability and peptide protonation constants (4.83 and 4.7 for CuHL; 5.36 and 5.56 for CuL, respectively) (Table 1). The remaining three species (CuH_−1_L, CuH_−2_L, CuH_−3_L) are due to the coordination of three amides (p*K*_a_ = 6.53, 6.48, 10.04, respectively). The CuH_−1_L complex, which has the first amide bond to the Cu(II), starts to form at pH around 6 and reaches its maximum at pH 6.5. This form is highly overlapped with the CuH_−2_L complex, which dominates over a wide pH range (6.5–10) (Appendix A). In the absorption spectrum, a broad signal at pH 6 suggests the presence of forms with Cu(II) coordinated by two and three nitrogens (Figure 2C), which is in agreement with the EPR spectrum showing overlapped signals from different Cu(II) species at pH 6.5 (Appendix A) [17,18]. Already at pH 6, the CD spectrum has two d-d features at 515(+) and 607(-) nm that indicate the square-planar complex is starting to form at this pH (Figure 3C) [19]. Above pH 7, the absorption maximum around 555 nm indicates coordination by four nitrogen [17] (Figure 2C), confirmed by the EPR parameters A_‖_ = 200.0 G and g_‖_ = 2.201 (spectra at pH 7.5, 9 and 10) [18]. In the CD spectrum, the intensity of the peaks found at pH 6 increases over the pH range 7–11, confirming the presence of a square-planar complex. The third amide group starts to coordinate at pH above 8 (CuH_−3_L) and is the most abundant form at alkaline pH (Appendix A).

Of particular interest for the Cu(II)-SAAP6 complex is the formation of its square-planar species at significantly lower pH (about pH two units lower) than the analogous Cu(II) complexes of SAAP2 and SAAP3. This indicates that Cu(II) coordination by carboxylate(s) of the first and/or second residue of the peptide suppresses its Lewis acidity and competition with proton(s) for the amides that result in square planar N_3_O or N_4_ coordination to the N-terminus of the peptide [17,18,19,20,21,22].

#### 2.2.4. A Comparison of the Stability of Metal Binding Abilities

Competition diagrams were prepared, based on the stability constants calculated from the potentiometric measurements. These present the hypothetical situation in which equimolar amounts of all three peptides are mixed, enabling a comparison of peptide affinity for a given metal ion. The graph shows that the affinities for Zn(II) are rather comparable, with a slight predominance of SAAP3 (DDDDDDD) over SAAP2 (GDDDDDD) and over SAAP6 (GADDDDD). These minor differences in the stabilities of the Zn(II) complexes correlate with the number of aspartic acid residues in the sequence of the peptide, which can serve as metal-binding residues (Figure 4A). Moreover, electrostatic interactions between the negatively charged aspartic acid residues and the positively charged metal ions may provide an additional stabilizing effect. The difference in stability among the Cu(II) complexes is more pronounced, but again with SAAP3 (DDDDDDD) the most stable and SAAP6 (GADDDDD) the least stable (Figure 4B). In this case, however, the origin of the difference is more complex. In contrast to SAAP2 and SAAP3, an amide participates in Cu(II) binding at a substantially lower pH for SAAP6, which lacks an Asp in the first and second positions. We would have anticipated a higher SAAP6 affinity for Cu(II) as amide-bound square-planar Cu(II)-peptide species have 5- and 6-membered chelate rings and are expected to be more stable. To further understand the stability of these Zn(II) and Cu(II) peptide complexes, ITC studies were undertaken to determine the thermodynamics of metal-peptide complex formation [23].

### 2.3. Isothermal Titration Calorimetry

Calorimetric measurements of metal ions binding to ligands, peptides, and proteins provide the enthalpic and entropic origin of the free energy of complex formation (stability). However, ITC measures the *net* thermodynamics of binding and buffer competition with the peptide for the metal, and proton competition with the metal for the peptide must be considered to find the condition-independent thermodynamic values [24,25]. Cacodylate (Caco) was used as the buffer for ITC measurements because it has been reported that Caco does not bind Cu(II) and, therefore, does not compete with the peptide for this metal ion [26]. However, Caco does bind Zn(II) with a condition–independent binding constant (K_ZnB_) of 147.9 (log K_ZnB_ = 2.17) [27], which was included in the data analysis. Representative ITC data for Zn(II) titrated into SAAP2, SAAP3, and SAAP6 at 25℃ in 25 mM Caco buffer and the net thermodynamic signatures of binding are shown in Figure 5. The best-fit values for the stoichiometry (n_ITC_), stability constant (K_ITC)_, and change in enthalpy (ΔH_ITC_) were obtained with a one set of sites binding model. All three peptides bind Zn(II) with similar and moderate affinity and with similar net values for the change in enthalpy and change in entropy, denoted as -TΔS (Table 3). In all three cases, the binding is due to a net favorable change in entropy. These results show that all three peptides bind Zn(II) with similar thermodynamics. The experimental equilibrium constant (K_ITC_) includes two competing equilibria, buffer competition with the peptide for the metal and proton competition with the metal for the peptide [28]. This competition is quantified by Equation (1):
(1)
$$K_{ITC}=\frac{K}{\alpha_{proton}\cdot \alpha_{buffer}}$$

where *K* is the pH- and buffer-independent metal-peptide binding constant, *α_proton_* is a function of the pH and p*K*_a_’s of the peptide and *α_buffer_* is a function of the metal-buffer binding constant (K_MB_) and the concentration of the basic form of the buffer that coordinates the metal ion [29]. The p*K*_a_s of the peptides have been determined from the potentiometric titrations and are found in Table 1.

The *α_proton_* accounts for Zn(II) competition with protons for the N-terminal amine and was quantified by Equation (2):
(2)
$$\alpha_{proton}=1+K_{N-t}\left[H+\right]$$


The calculated *α_proton_* value for SAAP2, SAAP3, and SAAP6 is 22.9, 11.7, and 20.9, respectively.

The *α_buffer_* accounts for buffer competition with the peptide for the Zn(II) and were quantified by Equation (3):
(3)
$$\alpha_{buffer}=1+K_{ZnB}\left[Caco\right]$$

and is equal to 4.5. Based on this analysis, the pH- and buffer- independent binding constants for Zn(II) binding to SAAP2, SAAP3, and SAAP6 determined from these ITC data are 3.2 × 10^6^, 1.5 × 10^6^ and 5.2 × 10^6^, respectively. These values are in good agreement with those obtained from potentiometric titrations (1.9 × 10^6^, 2.4 × 10^6^, and 1.3 × 10^6^, respectively) and confirm a similar affinity of all three peptides for Zn(II).

Guided by the potentiometric results, pH 6.8 was chosen for the Cu(II) ITC measurements because CuH_−1_L is the predominant SAAP2 and SAAP3 species at this pH, where it is also maximal for SAAP6, although CuH_−2_L is prominent for this peptide at pH 6.8. The initial ITC data for Cu(II) titrations of the peptides under the same conditions as the Zn(II) titrations were significantly different, showing biphasic binding to SAAP2 and SAAP3 and weak binding to SAAP6 (Appendix A). However, Cu(II) binding is slower than Zn(II) binding and additional time is required between each aliquot to reach equilibrium, with representative titrations shown in Figure 6. The stoichiometry of the biphasic binding indicates an initial endothermic formation of 1:2 M:L species followed by an exothermic transition to the 1:1 species upon further addition of Cu(II). These results distinguish Cu(II) binding to SAAP2 and SAAP3, with Asp carboxylates in the first and/or second position, from its binding to SAAP6, which lacks this residue in these positions.

Reversing the titration direction (peptide into Cu(II) solution) shows a 1:1 binding for all three peptides (Appendix A) and these data have been fit to a one set of sites binding model, giving the values in Table 3, which also contains best-fit values for the 1:1 binding of Cu(II) to SAAP6 in the forward direction. The pH- and buffer-independent binding constant for Cu(II) is not readily determined with Equation (1), as it is for Zn(II), due to proton competition for the amide, as well as the N-terminal amine, at this pH. Since there is no buffer competition (α_buffer_ term) for Cu(II), the experimental values give the binding constant at pH 6.8, which is 6.1 × 10^5^, 3.2 × 10^6^, and 5.2 × 10^5^ for SAAP2, SAAP3, and SAAP6, respectively. The value for SAAP6 in the forward direction is 4.6 × 10^5^, showing an internal consistency of the method. By comparing these values to those obtained from potentiometry in the competition plots (Figure 4), an important consistency in quantifying Cu(II)-peptide stability between the methods is revealed. The relative populations of the Cu(II) complexes of the three peptides at pH 6.8 in Figure 4B match the ratio of the Cu(II) binding constants at this pH, with SAAP3 4–5 times larger than SAAP2, which is similar to SAAP6. A similar comparison can be made for Zn(II), whose buffer-independent binding constants for SAAP2, SAAP3, and SAAP6 at pH 7.4 are 1.4 × 10^5^, 1.3 × 10^5^, and 2.5 × 10^5^, respectively. The relative populations of the Zn(II) complexes of the three peptides at pH 7.4 in the competition plot (Figure 4A) match the ratio of Zn(II) binding constants at this pH, which are all within a factor of two. Previously, we have studied the role of Asp and His side chains in the Cu(II) and Zn(II) coordination chemistry of peptides [20,30]. Here, ITC measurements confirm that all three of these Asp-rich peptides bind Zn(II) with similar affinities. Further, ITC measurements show that Cu(II) binding to SAAP2 and SAAP3 are similar and differ from Cu(II) binding to SAAP6, confirming the importance of Asp residues in the first two positions for the coordination and thermodynamics of Cu(II) binding, as well as the pH at which the first amide starts to bind the Cu(II).

Taken together, the results obtained herein show that the SAAP2, SAAP3, and SAAP6 peptides bind Zn(II) with a similar affinity in a net entropically-driven process with carboxylate groups from Asp side chains, providing an anchoring site and the N-terminal amine participating in Zn(II) coordination at neutral pH (Figure 7). Cu(II) binding to the SAAP2, SAAP3, and SAAP6 peptides is similarly a net entropically-driven process. However, the N-terminal amine serves as an anchoring site and three amide nitrogens bind sequentially with increasing pH (Figure 8).

### 2.4. Membrane Disrupting Ability

To examine the membrane disrupting the ability of SAAPs and their Zn(II) and Cu(II) complexes, liposomal experiments were carried out, where liposomes filled with fluorescent carboxyfluorescein serve as a bacterial membrane model. In the case of membrane (liposome) damage, dye leakage would have been visible in the fluorescence spectrum. During these experiments with SAAP and SAAP-metal complexes, no leakage was observed (data not shown).

These results are in agreement with previous suggestions that the antimicrobial activity of SAAP’s is due to intracellular targeting, not membrane impairment, consistent with results of Brogden et al. for mainly organelle distortion without cell membrane disruption [4,5]. Moreover, these results show that the presence of metal ions does not shift the SAAP mode of action into a membrane disrupting one.

### 2.5. Antimicrobial Activity

The need for the development of novel antimicrobial agents is well-known and well-justified. Numerous antimicrobials that were once capable of treating infections are now losing their utility and antimicrobial peptides may be worthy successors, as they possess a few advantages over conventional antimicrobial agents. Most important, despite the existence of AMPs in Nature for thousands of years, microbes have developed almost no resistance against them [32,33]. Among AMPs, anionic peptides (APs) are a relevant group of aspartic-acid-rich molecules that are generally isolated from mammalian epithelia [34]. SAAP2, SAAP3, and SAAP6 are proteins present in ovine surfactant extracts, bronchoalveolar lavage fluid, and airway epithelial cells. It was reported that they appear in millimolar concentrations and are antimicrobial against several microorganisms. As already mentioned, zinc as a cofactor is required for its antimicrobial activity [35]. In this work, we tested the antimicrobial activity of SAAP and SAAP-metal complexes on species other than *Mannheimia haemolytica* using an expanded panel of microorganisms: *Enterococcus faecium* BAA−2317, *Staphylococcus aureus* 43300, *Klebsiella pneumoniae* 700603, *Acinetobacter baumannii* 19606, *Pseudomonas aeruginosa* 27853, *Escherichia coli* 25922 and *Candida albicans* 10231.

In the studies of Brogden et al. [4,6] all three compounds were tested only against *Mannheimia haemolytica*, whereas other microorganisms were incubated only with SAAP3. Here we treated all microbiological strains with all three peptides with or without zinc(II) or copper(II) ions in the test buffer. The broth microdilution method was used to determine the MIC, and a modified Richard’s method was used to define MBC/MFC. Neither MIC nor MBC/MFC could be quantified in the tested concentration range (0.5–256 µg/mL) for any of the peptides and their complexes with Zn(II) or Cu(II) ions. This is consistent with the results of Brogden et al., which clearly indicates that MBC_50_ goes beyond our tested concentration range for the tested microorganisms, with the exception for *M. haemolytica*, whose MBC_50_ values for SAAP2, SAAP3, and SAAP6 in the presence of zinc ions are as low as 0.04, 0.02 and 0.02 mM (30.62, 16.47 and 14.43 μg/mL), respectively [4]. Here we demonstrate that the applicability of all three SAAP peptides is questionable for microbiological strains other than *M. haemolytica*. The use of an equimolar concentration of zinc(II) or copper(II) ions in the test buffer did not reveal any antimicrobial concentration of SAAP peptides lower than 256 µg/mL. Therefore, SAAP antimicrobial peptides seem to be very promising agents against *M. haemolytica* but not the human pathogens tested in this work.

## 3. Materials and Methods

All peptides (GDDDDDD, DDDDDDD, GADDDDD) were purchased from KareBay Biochem (USA) and were used as received (certified purity—98%). Carbonate-free stock solutions of 0.1 M NaOH were purchased from Sigma-Aldrich and then potentiometrically standardized with potassium phthalate (99.9% purity). Ethylene glycol and chloroform were bought from Chempur (pure p.a.). Methanol used for mass spectroscopy was purchased from J.T. Baker (LC-MS). NaClO_4_, Cu(ClO_4_)_2_, Zn(ClO_4_)_2_, NaCl, cholesterol, Triton X100, Sephadex G-50 and 6-carboxyfluorescein were bought from Sigma Aldrich. HClO_4_ was ordered from VWR. Lipids used for liposome experiments (DOPC, DOPA) were purchased from Avanti Polar Lipids. For antimicrobial susceptibility testing, tryptic soy broth (TSB) was acquired from Oxoid, whereas 2,3,5-triphenyltetrazolium chloride (TTC) was purchased from Alfa Aesar.

### 3.1. Mass Spectrometry

High-resolution mass spectra were obtained on a Bruker Apex Ultra FT-ICR (Bruker Daltonik, Bremen, Germany), equipped with an Apollo II electrospray ionization source with an ion funnel. The mass spectrometer was operated in the positive ion mode. The instrumental parameters were: scan range, *m/z* 100–1900; dry gas, nitrogen; temperature, 453 K; and ion energy, 5 eV. The capillary voltage was optimized to the highest S/N ratio, which was 4800 V. Samples were prepared in a 1:1 methanol–water mixture with a 1:1 M(II):peptide molar ratio, [peptide] = 0.1 mM, pH 7.4. The samples were infused at a flow rate of 3 μL min^−1^. The instrument was calibrated externally with a Tunemix^TM^ mixture (Bruker Daltonik, Germany) in quadratic regression model. Data were processed using the Bruker Compass DataAnalysis 4.0 program. The mass accuracy for the calibration was better than 5 ppm, which enabled, together with the true isotopic pattern (using SigmaFit), an unambiguous confirmation of the elemental composition of the complex.

### 3.2. Potentiometry

Stability constants for proton, Zn(II), and Cu(II) complexes of the peptides were calculated from pH-metric titration curves obtained over the pH range 2–11 at T = 298 K in a water solution with 4 mM HClO_4_ and 0.1 M NaClO_4_, using a total volume of 3 mL. The potentiometric titrations were performed using a Metrohm Titrando 905 titrator and a Mettler Toledo InLab Micro combined pH electrode. The thermostabilized glass cell was equipped with a magnetic stirring system, a micro burette delivery tube, and an inlet–outlet tube for argon. Solutions were titrated with 0.1 M carbonate-free NaOH. The electrodes were calibrated daily for hydrogen ion concentration by titrating HClO_4_ with NaOH under the same experimental conditions as above. The purities and exact concentrations of the peptide solutions were determined by the Gran method [36]. The peptide concentration was 0.5 mM and the Zn(II)- and Cu(II)-to-peptide ratio was 1:1.

The standard potential and the slope of the electrode couple were computed by means of the GLEE program [37]. The HYPERQUAD 2006 [16] program was used for the stability constant calculations. The standard deviations were computed by HYPERQUAD 2006 and refer to random errors only. The constants for the hydrolytic Zn(II) species were used in these calculations. The distribution and competition diagrams were computed with the HYSS program [38].

### 3.3. Absorption, Circular Dichroism, and Electron Paramagnetic Resonance Spectrometry

Absorption spectra were recorded on a Varian Cary300 Bio spectrophotometer and circular dichroism (CD) measurements were obtained on a Jasco J-1500 CD spectrometer. For both methods, spectra were collected over the 200–800 nm range using quartz cuvettes with an optical path of 1 cm. The solutions were prepared in a water solution containing 4 mM HClO_4_ and 0.1 M NaClO_4_. The concentrations of solutions used for spectroscopic studies were similar to those used for the potentiometric measurements with a 1:1 Cu(II)-to-peptide ratio. The absorption and CD spectroscopic parameters are from spectra obtained at pH values corresponding to the maximum concentration of each particular species, based on the distribution diagrams.

Electron paramagnetic resonance (EPR) spectra were recorded at 77 K at an X-band frequency (9.5 GHz) on a Bruker ELEXSYS E500 CW-EPR spectrometer equipped with an ER 036TM NMR Teslameter and E41 FC frequency counter. The peptides were prepared in an aqueous solution containing 4 mM HClO_4_ and 0.1 M NaClO_4_, with 30% ethylene glycol as a cryoprotectant. The concentration of Cu(II) was 1 mM with a 1:1 metal-to-peptide molar ratio. The pH was adjusted with appropriate amounts of HClO_4_ and NaOH solutions. The EPR parameters were determined from computer simulations of the experimental spectra using Bruker’s WIN-EPR SIMFONIA Software, Version 1.2 (Billerica, USA). All spectra were drawn in Origin 7.

### 3.4. Isothermal Titration Calorimetry

Isothermal titration calorimetry (ITC) measurements were carried out at 25 °C with a MicroCal PEAQ ITC. All reagents were >99% pure and obtained from Sigma-Aldrich. The peptides were dissolved directly into a 25 mM cacodylate (Caco, dimethylarsonate) buffer solution, whose pH was adjusted with 0.3 M NaOH and 0.3 M HClO_4_ to 7.4 for studies of zinc-binding and 6.8 for studies of copper binding. Metal stock solutions (copper(II) nitrate trihydrate, zinc(II) nitrate hexahydrate) were prepared in deionized water (18 MΩ) at low pH (~2) in acid-washed glass bottles and used to prepare 25 mM Caco buffer solutions, whose concentration was verified by ITC titrations with a standardized EDTA solution. After stabilizing the instrument at 25℃, 40 µL of a metal buffer solution (1.8–2 mM) was used to titrate 200 µL of a peptide buffer solution, whose concentration was initially ten times smaller than that of the metal ion. Each titration consisted of 19 successive injections with an interval of 180 s between each aliquot and a stirring speed of 750 rpm, which was repeated few times. For Cu(II), additional titrations were done with a longer spacing between injections to ensure that all of the heat associated with each aliquot was measured. The heat of a dilution from a corresponding control titration was subtracted before data fitting. An initial 0.4 µL injection was discarded from each data set to remove the effect of titrant diffusion across the syringe tip during the equilibration process. A CaCl_2_–EDTA titration was performed periodically for comparison to results obtained during the initial calibration of the instrument. The data were processed with MicroCal PEAQ-ITC Analysis Software. The upper panel of Figure 5 shows the heat flow with time during the titration and the lower panel (Figure 5) displays the integrated, concentration-normalized enthalpy for each aliquot versus the molar ratio of the titrant (metal) to the titrand (peptide) in the reaction cell. The solid line in the lower panel (Figure 5) indicates the best fit to a binding model.

### 3.5. Liposome Preparation and Leakage Assay

Liposomes were prepared by the methods of Jimah et al. [39] and Kacprzyk et al. [40] Ten mg of lipids (DOPC:DOPA:Cholesterol 30:30:40 mol%) were dissolved in 1 mL of chloroform and then dried under a stream of air for 4 h. The dry lipid film was hydrated with a solution of 6-carboxyfluorescein in HEPES buffer (20 mM 6-carboxyfluorescein, 10 mM HEPES, 150 mM NaCl, pH 7.4), sonicated, and subjected to four freeze-thaw cycles (−15 °C to 60 °C).

To obtain homogenous liposomes, the liposomal suspension was extruded 70 times through polycarbonate membranes with a pore size of 100 nm, mounted on Avanti Mini-Extruder (Avanti Polar Lipids). To remove untrapped 6-carboxyfluorescein, the liposomal suspension was filtered twice through a Sephadex-G50 column.

1 mM solutions of peptide or metal-peptide complex in HEPES buffer (10 mM HEPES, 150 mM NaCl, pH 7.4) were prepared. For the complexes, the metal-to-peptide ratio was 1:1. Each sample contained 1970 μL of HEPES buffer, 20 μL of liposomes, and 10 μL of the peptide or complex solution (the final concentration of peptide or complex was 5 μM). To obtain results for 100% dye leakage, 40 µL of 1% Triton X100 was added to the sample after each measurement. 1980 μL of HEPES buffer and 20 μL of liposomes were measured to check the passive dye leakage. Each experiment was carried out for 10 min with excitation and emission wavelengths of 492 nm and 512 nm, respectively.

Fluorescence spectra were recorded in the kinetic mode on a Varian Cary Eclipse spectrofluorometer using a quartz fluorescence cuvette with an optical path of 1 cm.

To assess the percent of disrupted liposomes, results were analyzed with the following formula:
(4)
$$Disruption_{time}=\left[\left(F_{10}-F_{0}\right)/\left(F_{100\%}-F_{0}\right)\right]\times 100\%$$

where *F*_10_ is the intensity of fluorescence after 10 min, *F*_0_ is the intensity of fluorescence in t = 0 s and *F*_100%_ is the intensity of fluorescence after Triton X100 addition [39,40].

### 3.6. Antibacterial Activity

Seven reference strains from ATCC (Enterococcus faecium BAA-2317, Staphylococcus aureus 43300, Klebsiella pneumoniae 700603, Acinetobacter baumannii 19606, Pseudomonas aeruginosa 27853, Escherichia coli 25922, and Candida albicans 10231) were used for antimicrobial activity assays. The minimum inhibitory concentration (MIC) was determined by a microdilution method with spectrophotometric measurements according to ISO 20776-1:2019 [41] and ISO 16256:2012 [42], whereas a modified Richard’s method using the redox indicator (2,3,5-triphenyltetrazolium chloride, TTC) was used to determine the minimal bactericidal/fungicidal concentration (MBC/MFC) [43,44]. Briefly, two-fold serial dilutions of the peptides in tryptic soy broth (TSB), with or without an equimolar concentration of Zn(II) or Cu(II), were prepared in a 96-well microplate (final concentrations ranged from 0.5 to 256 µg/mL) and then incubated with bacterial or fungal suspension (about 5 × 10^5^ CFU/mL) at 37 ± 1 °C (bacteria strains) or 25 ± 1 °C (fungus strain) for 24 h on the shaker (500 rpm). Negative (TSB + strain) and medium (only TSB) controls were also included, as well as a spectrophotometric solubility control for each peptide or metal-peptide system. To avoid using strains that had developed additional resistance, the following antibacterial/antifungal agents were used as a positive control, according to breakpoint values established by the EUCAST [45]: E. faecium: 4 µg/mL ampicillin, S. aureus: 1 µg/mL levofloxacin, K. pneumoniae: 2 µg/mL gentamicin, *A. baumannii*: 0.5 µg/mL levofloxacin, P. aeruginosa: 1 µg/mL levofloxacin, E. coli: 2 µg/mL gentamicin, C. albicans: 1 µg/mL amphotericin B. After incubation, spectrophotometric measurements were performed at 580 nm, after which a 50 µL aliquot of 1% (*m/v*) TTC solution was added to each well. The MIC was the lowest peptide concentration causing a 50% loss of viability. The MBC/MFC was the lowest concentration that did not show microbial growth by visual analysis after 24 h incubation with TTC, as indicated by the lack of enzymatic reduction to red 1,3,5-triphenylformazan (TPF).

## 4. Conclusions

The binding of Zn(II) and Cu(II) to SAAP2, SAAP3, and SAAP6 are entropically driven processes. Zn(II) has a similar affinity for the studied peptides, while the stability of the Cu(II)-peptide complexes increase with the number of Asp residues—when these carboxylates are in the first and/or second positions, they play an important role in the Cu(II) coordination. SAAP6, which lacks an Asp in the first two positions, exhibits amide coordination of the Cu(II) at a much lower pH than the other two peptides. Clearly, Cu(II) coordination by the N-terminal amine and carboxylates in the first and/or second positions modulates its Lewis acidity for competition with amide protons and its peptide affinity. Interestingly, the amide coordination by SAAP6 does not lead to a higher affinity for the Cu(II) ion. Taken together, we can point to an important role for the first two amino acids in the SAAP sequence for binding Cu(II), as well as determining the pH at which the first amide starts to participate in Cu(II) binding.

As for the SAAP antimicrobial mode of action, neither the peptides nor their Zn(II) or Cu(II) complexes damages the cell membrane, which was confirmed by liposomal experiments. No significant antimicrobial activity (MIC < 256 µg/mL) was observed against the human pathogens studied in this work, making SAAP peptides and their metal complexes promising only for anti-*M. haemolytica* treatment. Still unclear, however, is the molecular origin of the antimicrobial activity against *M. haemolytica* in the presence of Zn(II) [6]. The effect of charge seems unlikely, as Zn(II) binding at neutral pH only reduces the negative charge of the peptide by one as Zn(II) binds to the N-terminal amine and displaces its proton. Perhaps it is the structure or conformation of the Zn(II)-peptide complex, such as that shown in Figure 7, that enhances the antimicrobial properties of the SAAP peptides.

## Figures and Tables

**Figure 1 ijms-22-06971-f001:**
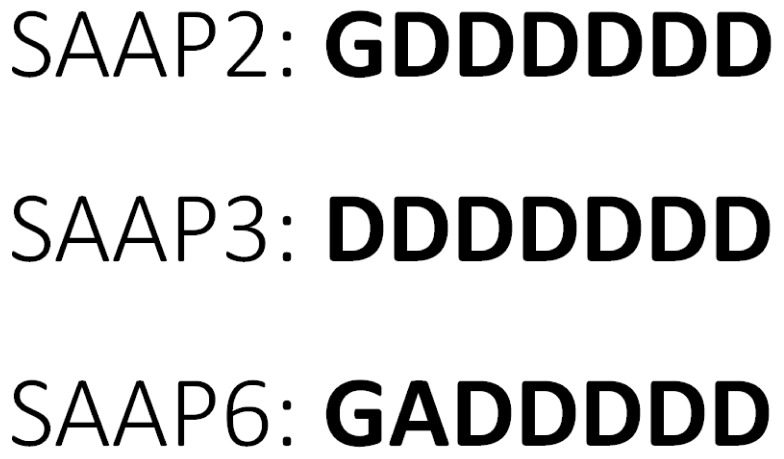
Amino acid sequences of SAAP2, SAAP3, and SAAP6 peptides.

**Figure 2 ijms-22-06971-f002:**
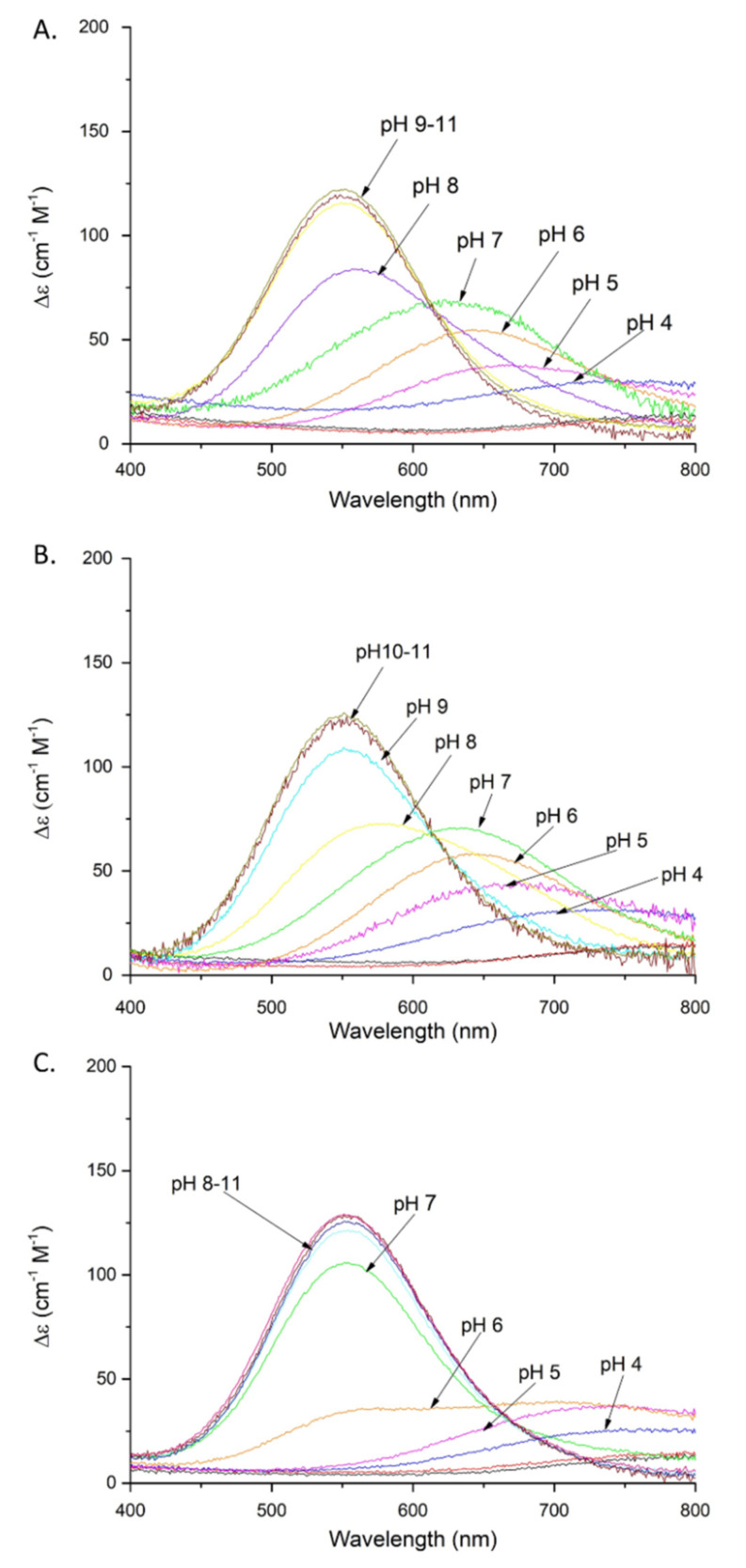
Absorption spectra of Cu(II) complexes with: (**A**) SAAP2; (**B**) SAAP3; (**C**) SAAP6; over the pH range 2–11. Conditions: T = 298 K, 0.1 M NaClO_4_, [Cu(II)] = [SAAP2] = [SAAP3] = [SAAP6] = 1.0 mM.

**Figure 3 ijms-22-06971-f003:**
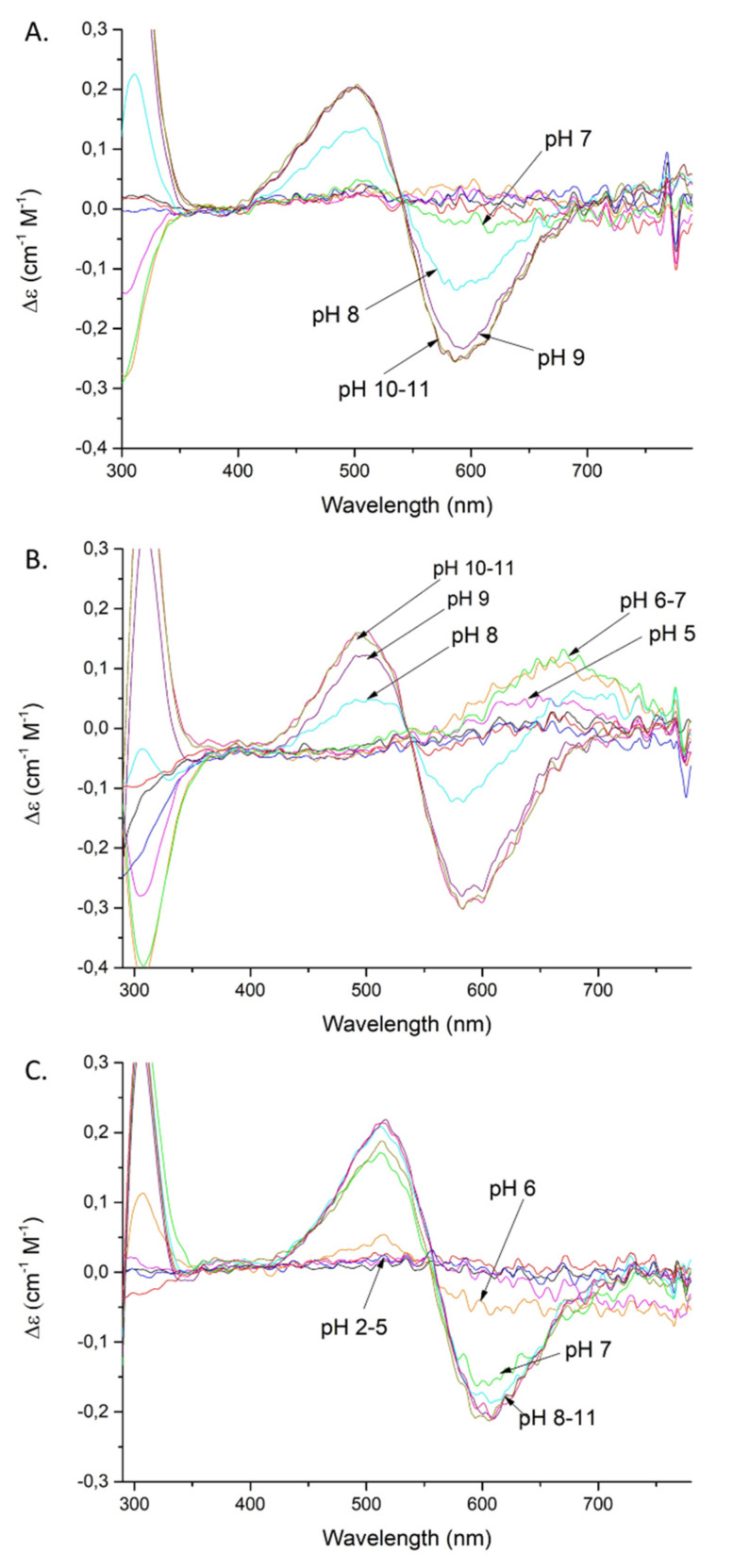
CD spectra of Cu(II) complexes with: SAAP2 (**A**); SAAP3 (**B**); SAAP6 (**C**); over the pH range 2–11. Conditions: T = 298 K, 0.1 M NaClO*_4_*, [Cu(II)] = [SAAP2] = [SAAP3] = [SAAP6] = 1.0 mM.

**Figure 4 ijms-22-06971-f004:**
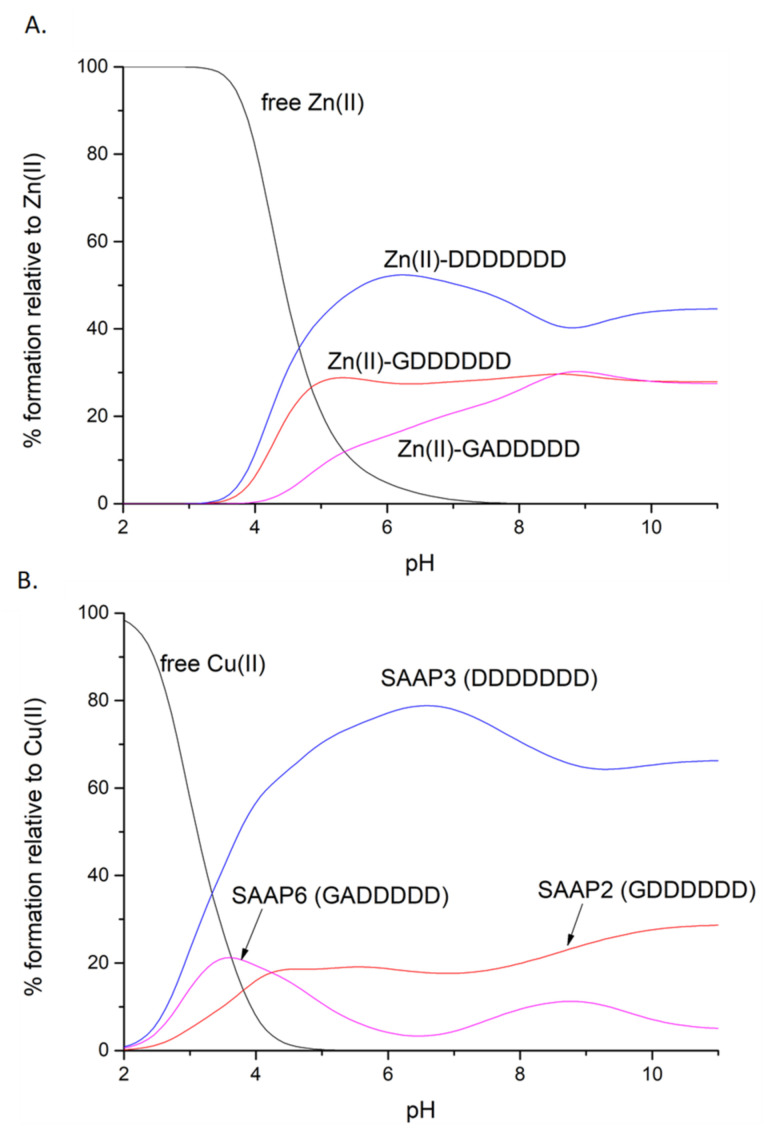
Competition plots for SAAP2, SAAP3, SAAP6, and Zn(II) (**A**) or Cu(II) (**B**), showing the relative amount of each complex at different pH values for the hypothetical situation in which equimolar amounts of the four species are mixed.

**Figure 5 ijms-22-06971-f005:**
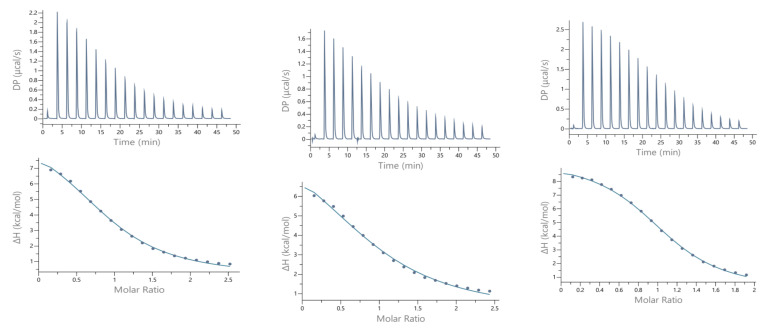
Representative ITC data (above) and their corresponding thermodynamic signatures (below) for Zn(II) (1.8–2.0 mM) titrated into (**A**) SAAP2, (**B**) SAAP3 and (**C**) SAAP6 peptides (140–180 µM) at pH 7.4.

**Figure 6 ijms-22-06971-f006:**
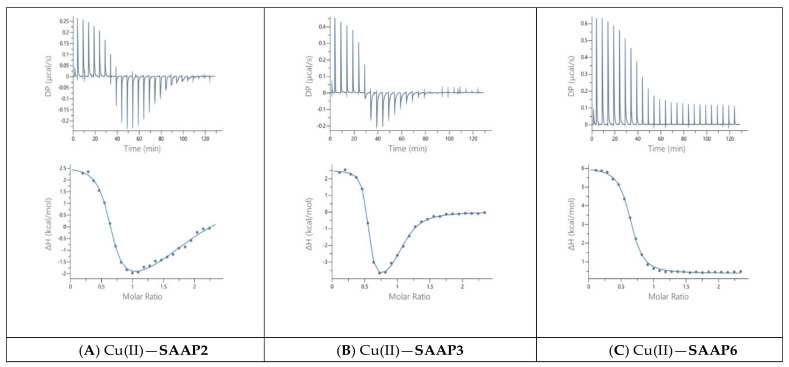
Representative ITC data for Cu(II) (1.8 mM) titrated into (**A**) SAAP2, (**B**) SAAP3, and (**C**) SAAP6 peptides (150 µM) at pH 6.8.

**Figure 7 ijms-22-06971-f007:**
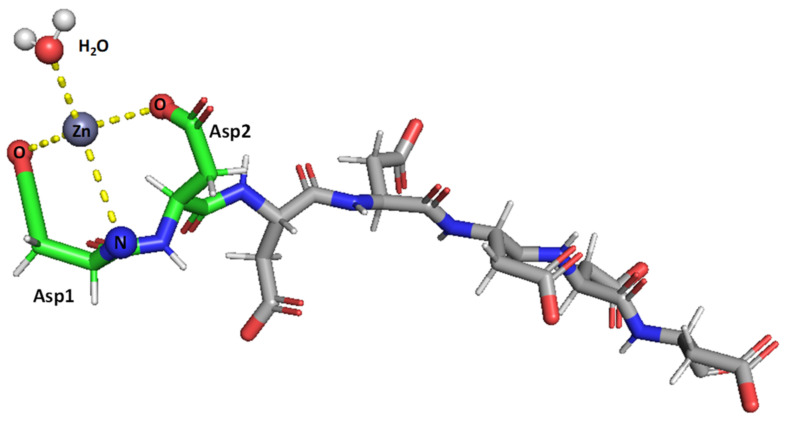
Suggested mode of coordination for Zn(II)-SAAP3 at pH 8. Zn(II) is bound by two carboxylates from aspartic acid side chains and by the N-terminal amine. A water molecule is also bound to create tetrahedral coordination. Figures were generated using PyMOL [31].

**Figure 8 ijms-22-06971-f008:**
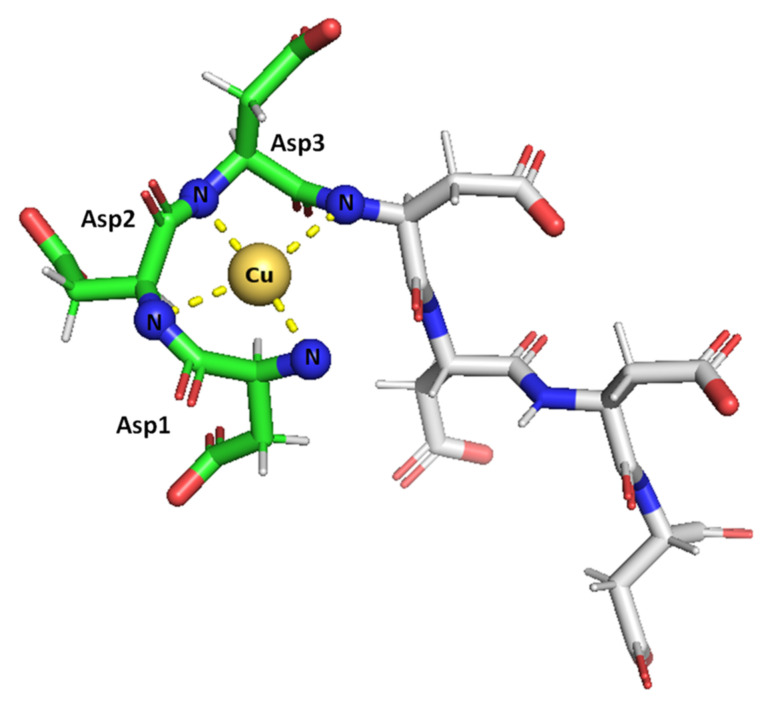
Suggested mode of coordination for Cu(II)-SAAP3 at pH 9. Cu(II) is bound by the N-terminal amine and three amide nitrogens, giving square-planar coordination. Figures were generated using PyMOL [31].

**Table 1 ijms-22-06971-t001:** Potentiometric data for proton, Zn(II) and Cu(II) complexes with SAAP2 (GDDDDDD), SAAP3 (DDDDDDD) and SAAP6 (GADDDDD). Titrations were carried out over the pH range 2–11 at T = 298 K in an aqueous solution with 4 mM HClO_4_ and 0.1 M NaClO_4_. The peptide concentration was 0.5 mM and the Zn(II)- and Cu(II)-to-peptide ratio was 1:1. HYPERQUAD 2006 [16] was used to determine the stability constants. Standard deviations are shown in brackets. N-t refers to the N-terminal amine group.

	SAAP2 (GDDDDDD)		SAAP3 (DDDDDDD)		SAAP6 (GADDDDD)	
species	logβ	logK	logβ	logK	logβ	logK
LH	8.74 (4)	8.74 (N-t)	8.43 (2)	8.43 (N-t)	8.70 (1)	8.70 (N-t)
LH_2_	14.30 (8)	5.56 (Asp)	14.32 (4)	5.89 (Asp)	14.26 (3)	5.56 (Asp)
LH_3_	18.99 (8)	4.69 (Asp)	19.30 (3)	4.98 (Asp)	18.96 (2)	4.70 (Asp)
LH_4_	23.27 (10)	4.28 (Asp)	23.89 (4)	4.59 (Asp)	23.19 (3)	4.23 (Asp)
LH_5_	26.89 (10)	3.62 (Asp)	28.02 (3)	4.13 (Asp)	26.73 (2)	3.54 (Asp)
LH_6_	29.89 (12)	3.00 (Asp)	31.52 (4)	3.50 (Asp)	29.70 (2)	2.97 (Asp)
LH_7_			34.72 (3)	3.20 (Asp)		
Zn(II) complexes						
ZnH_3_L			22.49 (4)			
ZnH_2_L	17.44 (4)		17.84 (3)	4.65 (Asp)		
ZnHL	12.58 (2)	4.86 (Asp)	12.92 (2)	4.92 (Asp)	12.15 (4)	
ZnL	6.27 (3)	6.31 (N-t)	6.38 (2)	6.54 (N-t)	6.10 (3)	6.05 (N-t)
ZnH_−1_L	−1.97 (3)	8.24 (H_2_O)	−1.96 (2)	8.34 (H_2_O)	−1.91 (4)	8.01 (H_2_O)
ZnH_−2_L	−10.91 (3)	8.94 (H_2_O)	−10.74 (2)	8.78 (H_2_O)	−10.86 (4)	8.95 (H_2_O)
ZnH−_3_L	−20.26 (3)	9.35 (H_2_O)	−19.88 (2)	9.14 (H_2_O)	−20.21 (4)	9.35 (H_2_O)
Cu(II) complexes						
CuH_5_L			31.68 (3)			
CuH_4_L	26.19 (5)		-		26.53 (7)	
CuH_3_L	-		24.40 (2)		-	
CuH_2_L	18.81 (2)		19.87 (3)	4.53 (Asp)	18.75 (5)	
CuHL	14.09 (3)	4.72 (Asp)	15.46 (3)	4.41 (Asp)	13.92 (5)	4.83 (Asp)
CuL	9.22 (2)	4.87 (Asp)	10.13 (4)	5.33 (Asp)	8.56 (6)	5.36 (Asp)
CuH_−1_L	3.20 (2)	6.02 (amide)	4.23 (5)	5.90 (amide)	2.03 (6)	6.53 (amide)
CuH_−2_L	−4.12 (3)	7.32 (amide)	−3.47 (7)	7.70 (amide)	−4.45 (8)	6.48 (amide)
CuH−_3_L	−13.59 (4)	9.47 (amide)	−12.89 (10)	9.42 (amide)	−14.49 (12)	10.04 (amide)

**Table 2 ijms-22-06971-t002:** EPR spectral parameters at different pH values obtained from X-band EPR spectra of frozen solutions (77 K) of 1 mM Cu(II)-SAAP2, Cu(II)-SAAP3, and Cu(II)-SAAP6 (1:1 Cu(II)-to-peptide molar ratio). Samples were in aqueous solutions with 4 mM HClO_4_ and 0.1 M NaClO_4_ and 30% ethylene glycol. Parameters marked in gray at pH 6.5 correspond to two overlapped different complex species.

	Cu(II)-SAAP2				Cu(II)-SAAP3				Cu(II)-SAAP6			
	A_‖_ [G] (A*_zz_*)	g_‖_ (g*_z_*)	g_⊥_ (g*_x_ =* g*_y_*)	MW Frequency [GHz]	A_‖_ [G] (A*_zz_*)	g_‖_ (g*_z_*)	g_⊥_ (g*_x_ =* g*_y_*)	MW Frequency [GHz]	A_‖_ [G] (A*_zz_*)	g_‖_ (g*_z_*)	g_⊥_ (g*_x_ =* g*_y_*)	MW Frequency [GHz]
3	121.3	2.412	2.080	9.5788	121.4	2.418	2.080	9.5735	121.3	2.410	2.080	9.5873
4	140.6	2.370	2.070	9.5884	140.4	2.370	2.065	9.5758	141.7	2.364	2.074	9.5884
5	140.0	2.365	2.071	9.5857	140.0	2.367	2.065	9.5793	137.5	2.364	2.065	9.5857
6.5	133.5	2.372	2.056	9.5878	131.5	2.375	2.062	9.5778	133.5	2.371	2.056	9.5878
	180.0	2.251	2.055		180.0	2.258	2.050		200.0	2.202	2.045	
7.5	179.0	2.252	2.050	9.5827	182.0	2.247	2.047	9.5784	200.0	2.201	2.043	9.5832
9	200.2	2.205	2.040	9.5802	206.0	2.201	2.045	9.5741	204.3	2.202	2.043	9.5840
10	200.2	2.205	2.040	9.5762	206.0	2.201	2.045	9.5778	204.3	2.202	2.045	9.5888

**Table 3 ijms-22-06971-t003:** Experimental thermodynamic values for Zn(II) and Cu(II) binding to SAAP2, SAAP3, and SAAP6 from ITC measurements in 25 mM cacodylate buffer at 25 °C.

Ligand	K_dITC_ [M]	ΔH_ITC_ [kcal/mol]	-TΔS_ITC_ [kcal/mol]	n_ITC_
Binding to Zn(II) (pH 7.4)				
**SAAP2**	(32 ± 2) × 10^−6^	8.7 ± 0.2	−14.8	0.92 ± 0.01
**SAAP3**	(35 ± 2) × 10^−6^	7.6 ± 0.2	−13.7	0.96 ± 0.01
**SAAP6**	(18.1 ± 0.6) × 10^−6^	9.4 ± 0.1	−15.9	1.05 ± 0.01
Binding to Cu(II) (pH 6.8)				
**SAAP2 (r)**	(1.6 ± 0.2) × 10^−6^	4.6 ± 0.1	−12.5	1.20 ± 0.01
**SAAP3 (r)**	(0.31 ± 0.06) × 10^−6^	4.2 ± 0.1	−13.0	1.29 ± 0.01
**SAAP6 (r)**	(1.9 ± 0.3) × 10^−6^	7.2 ± 0.2	−15.0	1.02 ± 0.01
**SAAP6 (f)**	(2.2 ± 0.2) × 10^−6^	6.0 ± 0.1	−13.7	0.61 ± 0.01

## Data Availability

All the data supporting the conclusions of this article are provided within the article and in its additional files. All data and materials are available upon reasonable request from the corresponding authors.

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
