# Peer review of "Zn-Enhanced Asp-Rich Antimicrobial Peptides: N-Terminal Coordination by Zn(II) and Cu(II), Which Distinguishes Cu(II) Binding to Different Peptides"

_ijms, 2021, doi:10.3390/ijms22136971_

Round 1

Reviewer 1 Report

Dear Dr. Spasojevic,

please find here my comments on the manuscript entitled “Zn-enhanced Asp-rich antimicrobial peptides: N-terminal coordination by Zn(II) and Cu(II), which distinguishes Cu(II) binding to different peptides” submitted for consideration for publication on IJMS as a full paper.

The paper deals with the characterization in solution of three Asp-rich oligopeptides that belong to the family of surfactant-associated anionic peptides. These three peptides have previously showed antbacterial properties against M. haemolytica. Here, in the broader context of the research carried out by the authors’ research group on metallopeptides with bactericidal activity, the three peptides and the formation of their adducts with Cu(II) and Zn(II) were studied using potentiometry, spectroscopic and spectrometric techniques, and ITC. The antimicrobial properties of these peptides and their metal adducts against a broader spectrum of fungi and bacteria, and their capacity to interact with liposome membranes are also reported.

Overall this is a rather complete study that has in the speciation in solution of these systems the starting point to understand the antimicrobial role of these peptides. Although it confirms that these pepetides are active only against M. haemolytica, therefore assessing a bacterium-dependent activity, the study of these peptides rich in aspartate is certainly interesting for the community of bioinorganic chemistry. For these reasons I would recommend the acceptance of this paper, after minor revisions.

- Page 10 Column 1 Line 12 from bottom: “The stoichiometry of the biphasic binding indicates an initial endothermic formation of a 1:2 dimer…” Using the term “dimer” can be confusing. It’s just a 1:2 M:L species.

- On a separate point, these 1:2 species could in theory have been seen also by potentiomery, provided the experiment is run in excess of ligand. The authors explored 1:1 Cu/L conditions only (ad far as it is understood from the text). This does not change much in the interpretation of the results, since all other experiments were run under a 1:1 metal/ligand stoichiometry.  It is just a pity that the data for those ratio are missing, since a complete comparison between potentiometric and ITC data could be made.

- Figure 7. I would move this at the end of potentiometry, not in the conclusion section. Also, figures come from PyMol, but I guess the models were generated previously using a different software. If so, this should be indicated.

Author Response

Dear Editor, dear Reviewer,

Thank you for considering our manuscript to be published in the International Journal of Molecular Sciences, and thank you for your detailed comments on our work. We have followed the Reviewers’ advice and made changes in our work, which are described below:

Reviewer 1 

The paper deals with the characterization in solution of three Asp-rich oligopeptides that belong to the family of surfactant-associated anionic peptides. These three peptides have previously showed antbacterial properties against M. haemolytica. Here, in the broader context of the research carried out by the authors' research group on metallopeptides with bactericidal activity, the three peptides and the formation of their adducts with Cu(II) and Zn(II) were studied using potentiometry, spectroscopic and spectrometric techniques, and ITC. The antimicrobial properties of these peptides and their metal adducts against a broader spectrum of fungi and bacteria, and their capacity to interact with liposome membranes are also reported. 

Overall this is a rather complete study that has in the speciation in solution of these systems the starting point to understand the antimicrobial role of these peptides. Although it confirms that these pepetides are active only against M. haemolytica, therefore assessing a bacterium-dependent activity, the study of these peptides rich in aspartate is certainly interesting for the community of bioinorganic chemistry. For these reasons I would recommend the acceptance of this paper, after minor revisions. 

- Page 10 Column 1 Line 12 from bottom: "The stoichiometry of the biphasic binding indicates an initial endothermic formation of a 1:2 dimer..." Using the term "dimer" can be confusing. It's just a 1:2 M:L species. 

We corrected this issue, thank you for pointing this out.

- On a separate point, these 1:2 species could in theory have been seen also by potentiomery, provided the experiment is run in excess of ligand. The authors explored 1:1 Cu/L conditions only (ad far as it is understood from the text). This does not change much in the interpretation of the results, since all other experiments were run under a 1:1 metal/ligand stoichiometry.  It is just a pity that the data for those ratio are missing, since a complete comparison between potentiometric and ITC data could be made. 

This is a valuable comment, to which it is not trivial to respond with a short answer. In principle it is true, that such a comparison would have been interesting, however, potentiometric and ITC results cannot always be directly compared to one another, mainly since they require different buffers which largely influence the binding constants. In the future, we will consider more detailed thermodynamic (ITC) studies, in which different buffers and different metal:ligand stoichiometries will be used.

- Figure 7. I would move this at the end of potentiometry, not in the conclusion section. Also, figures come from PyMol, but I guess the models were generated previously using a different software. If so, this should be indicated. 

We have move the figure (Figure 7 and Figure 8 as well) as advised. All the figures come from PyMOL only, no different software was previously used.

We would like to thank the Reviewer for the detailed comments. They really did help us to improve the manuscript.    

With our best regards,

Authors

Reviewer 2 Report

This study has utilized both calorimetry and ITC measurements to show that the unique Asp-only peptide SAAP3 (DDDDDDD) and its analogues SAAP2 (GDDDDDD) and SAAP6 (GADDDDD) have similar micromolar affinity for Zn(II). All the three peptides examined do bind Cu(II) with a net entropically-driven process but with higher affinity than they bind Zn(II) and a coordination that involves the N-terminal amine and deprotonated amides as the pH increases. It was then shown that both Zn and Cu ions do not stabilize a membrane disrupting ability and that they do not induce antimicrobial activity of these peptides against a panel of human pathogens. Although this study can be interesting to the readers of the journals, there are significant shortcomings that should be fixed before final acceptance of the ms. My comments are given below.

  1. Quality of pictures provided in the ms is too poor. High resolution pictures must be provided.
  2. Why are Figs 7 and 8 provided after conclusion section?
  3. Conclusion of the paper is too big. It should be reduced to a maximum of 10 lines.
  4. It is not clear why Zn complexes are entropically driven?
  5. Why are methodological details provide before the conclusion section? They should appear after the introduction section.
  6. There are typos and grammatical errors all over the ms. These should be corrected.
  7. Comparison of stability of constants obtained from this work should be made with similar systems reported previously.
  8. The paragraph “The Zn(II) and Cu(II) complexes of SAAP2, SAAP3 and SAAP6 were examined by a variety of analytical methods. Mass spectrometry indicates the stoichiometry of the complexes and also checks the purity of the samples. Potentiometric titrations determine the protonation constants and stability constants for distribution and competition diagrams that describe the stability of the complexes and suggest the mode of coordination. Isother-mal titration calorimetry data provided the thermodynamics of complex formation. Spectroscopic measurements indicate the number and type of donor atoms in the Cu(II) com-plexes, depending on the pH. Liposome experiments confirm that neither the peptidesnor their Cu(II) or Zn(II) complexes are membrane-disrupting molecules.” should appear in the introduction section. It has no place in the results and discussion section.

Author Response

Dear Editor, dear Reviewer,

Thank you for considering our manuscript to be published in the International Journal of Molecular Sciences, and thank you for your detailed comments on our work. We have followed the Reviewers’ advice and made major changes in our work, which are described below:

Reviewer 2 

This study has utilized both calorimetry and ITC measurements to show that the unique Asp-only peptide SAAP3 (DDDDDDD) and its analogues SAAP2 (GDDDDDD) and SAAP6 (GADDDDD) have similar micromolar affinity for Zn(II). All the three peptides examined do bind Cu(II) with a net entropically-driven process but with higher affinity than they bind Zn(II) and a coordination that involves the N-terminal amine and deprotonated amides as the pH increases. It was then shown that both Zn and Cu ions do not stabilize a membrane disrupting ability and that they do not induce antimicrobial activity of these peptides against a panel of human pathogens. Although this study can be interesting to the readers of the journals, there are significant shortcomings that should be fixed before final acceptance of the ms. My comments are given below. 

  1. Quality of pictures provided in the ms is too poor. High resolution pictures must be provided. 

We improved the quality of the pictures, thank you for pointing this out.

2.Why are Figs 7 and 8 provided after conclusion section? 

Figs. 7 and 8 illustrate what we write about in the conclusions section; we now moved them to the ‘results and discussion’ section.

3.Conclusion of the paper is too big. It should be reduced to a maximum of 10 lines. 

We did our best to shorten the conclusions section.

4.It is not clear why Zn complexes are entropically driven? 

From the chemical point of view, this behaviour could have been expected; the ITC data presented in the work confirm this with clear signatures of zinc binding. Most probably, the increase of entropy comes from liberation of water molecules upon Zn(II) binding, that surpasses the enthalpy of other processes.

5.Why are methodological details provide before the conclusion section? They should appear after the introduction section. 

This is a matter of the journal’s template, which includes the ‘1. Introduction, 2. Results, 3. Discussion, 4. Materials and Methods, 5. Conclusions’ scheme. This scheme is also observed in other papers from IJMS, e.g. https://www.mdpi.com/1422-0067/22/12/6414/htm, https://www.mdpi.com/1422-0067/22/12/6428/htm or https://www.mdpi.com/1422-0067/22/12/6416/htm.

6.There are typos and grammatical errors all over the ms. These should be corrected. 

We did our best to correct these errors, with the help of one of our co-authors, who is a native English speaker.

7.Comparison of stability of constants obtained from this work should be made with similar systems reported previously. 

Below, we compared the SAAP stability constants with the Cu(II) complex of the GGGG peptide (Figure 1) and with Zn(II) and Cu(II) complexes of a DDD peptide (Figure 2) on competition diagrams, which are based on the stability constants calculated from potentiometric measurements. The plots present a hypothetical situation in which equimolar amounts of all reagents are mixed, enabling a comparison of peptide affinity for a given metal ion. In the case of Cu(II), the graph shows that the presence of carboxylic side chains strongly enhances the peptides’ affinity for Cu(II), with the Cu-GGGG complex being much less stable than all Cu(II)-SAAP peptides (Figure 1). The stability of Cu(II)-DDD is quite comparable to that of SAAP2 and SAAP6 (Figure 2).

The affinity of Zn(II) for DDD is lower than that of longer Asp-rich peptides, as expected (Figure 3), most likely due to the fact that not all Zn(II) coordination sites are filled.

Figure 1 Competition plot for SAAP2, SAAP3, SAAP6, GGGG and Cu(II), showing the relative amount of each complex at different pH values for the hypothetical situation in which equimolar amounts of the five species are mixed. Constants for Cu(II)-GGGG taken from H. Sigel and R. B. Martin, Chem. Reo., 1982, 82, 385.

Figure 2. Competition plot for SAAP2, SAAP3, SAAP6, DDD and Cu(II), showing the relative amount of each complex at different pH values for the hypothetical situation in which equimolar amounts of the five species are mixed. Constants for Cu(II)-DDD taken from Kozlowski, H., Lebkiri, A., Onindo, C. O., Pettit, L. D., & Galey, J. F. (1995), Polyhedron, 14(2), 211-218.).

Figure 3. Competition plot for SAAP2, SAAP3, SAAP6, DDD and Zn(II), showing the relative amount of each complex at different pH values for the hypothetical situation in which equimolar amounts of the five species are mixed. Constants for Zn(II)-DDD taken from Kozlowski, H., Lebkiri, A., Onindo, C. O., Pettit, L. D., & Galey, J. F. (1995), Polyhedron, 14(2), 211-218.).

8.The paragraph "The Zn(II) and Cu(II) complexes of SAAP2, SAAP3 and SAAP6 were examined by a variety of analytical methods. Mass spectrometry indicates the stoichiometry of the complexes and also checks the purity of the samples. Potentiometric titrations determine the protonation constants and stability constants for distribution and competition diagrams that describe the stability of the complexes and suggest the mode of coordination. Isother-mal titration calorimetry data provided the thermodynamics of complex formation. Spectroscopic measurements indicate the number and type of donor atoms in the Cu(II) com-plexes, depending on the pH. Liposome experiments confirm that neither the peptidesnor their Cu(II) or Zn(II) complexes are membrane-disrupting molecules." should appear in the introduction section. It has no place in the results and discussion section. 

We have now moved the paragraph, as advised.

We would like to thank the Reviewer for the detailed comments. They really did help us to improve the manuscript.    

With our best regards,

Authors

Round 2

Reviewer 2 Report

Authors of this study have considered my comments, thus revised their paper. 

I suggest publication of this work.